# Test Method for Studying the Shrinkage Effect under Controlled Environmental Conditions for Concrete Reinforced with Coconut Fibres

**DOI:** 10.3390/ma16083247

**Published:** 2023-04-20

**Authors:** Mary Amaguaña, Leidy Guamán, Nicolay Bernardo Yanchapanta Gómez, Majid Khorami, María Calvo, Jorge Albuja-Sánchez

**Affiliations:** 1Departamento de Ingeniera Civil y Ambiental, Escuela Politécnica Nacional (EPN), Quito 170143, Ecuador; 2Architecture Department, Universidad Politécnica Salesiana (UPS), Quito 170525, Ecuador; 3Faculty of Engineering, Laboratory of Materials Resistance, Soil Mechanics, Pavements and Geotechnics, Pontificia Universidad Católica del Ecuador (PUCE), Quito 170143, Ecuador; 4International Faculty of Innovation PUCE-Icam, Pontificia Universidad Católica del Ecuador (PUCE), Quito 170143, Ecuador; 5Department of Engineering, University of Ferrara, 44122 Ferrara, Italy

**Keywords:** cracking behaviour, shrinkage, coconut fibre, wind tunnel

## Abstract

This study proposes a novel test method and corresponding procedure to evaluate how coconut fibres affect crack propagation rates resulting from plastic shrinkage during the accelerated drying of concrete slabs. The experiment employed concrete plate specimens, which were used to simulate slab structural elements with a surface dimension notably greater than their thickness. These slabs were reinforced with coconut fibre with 0.5%, 0.75%, and 1% fibre content. A wind tunnel was designed to simulate two significant climate parameters (wind speed and air temperature), which could impact the cracking behaviour of surface elements. The proposed wind tunnel allowed air temperature to be controlled alongside wind speed while monitoring moisture loss and the cracking propagation process. During testing, a photographic recording method was used to evaluate cracking behaviour, with the total crack length serving as a parameter to assess the impact of fibre content on the crack propagation of slab surfaces. Additionally, crack depth was measured using ultrasound equipment. The results indicate that the proposed test method was appropriate for future research, allowing for the evaluation of the effect of natural fibres on the plastic shrinkage behaviour of surface elements under controlled environmental conditions. Based on initial studies and the results obtained through the proposed test method, concrete containing 0.75% fibre content exhibited significantly reduced crack propagation on slab surfaces, as well as a reduction in the crack depth caused by plastic shrinkage during the early age of the concrete.

## 1. Introduction

The construction industry is constantly exploring new and innovative ways to create stronger and more durable materials while also reducing their impact on the environment. One such avenue of research is the use of natural fibres as reinforcements in concrete. In recent years, coconut fibres have emerged as a promising option due to their advantages [1].

Coconut fibres are versatile and sustainable materials that can be used to enhance the properties of concrete. One of the main advantages of coconut fibres is their high tensile strength, which makes them an exceptional material for controlling the cracks caused by the plastic shrinkage of the concrete when added to concrete [2,3,4]; these fibres help distribute the stresses more evenly throughout the material, which increases its overall tensile strength [5,6,7]. Coconut fibres have a low-cost material in construction. Compared to synthetic fibres and steel reinforcements, coconut fibres are a more affordable alternative, making them an attractive option for construction projects with tight budgets [8]. 

Coconut fibres are also an eco-friendly choice for construction projects. They are a renewable resource and are biodegradable, making them a sustainable alternative to synthetic fibres and steel reinforcement. Additionally, the use of coconut fibres in concrete can increase the durability of the material, as well as its resistance to cracking and shrinkage, which can help prolong the lifespan of the structure [9,10,11,12].

Nowadays, fibres are incorporated in concrete to produce materials with improved strength and toughness, such as coconut, bamboo, jute, palm, sisal, banana, sugarcane, abaca, and others which are flexible and easy to handle, especially when they are used in large quantities [13]. Some examples of the uses of natural fibres are: date palm fibres, which were used to control the influence of elevated temperatures, had a considerable impact on the longevity of concrete structures and improved the serviceability of structural elements [9,14], and the durability of concrete. On the other hand, is important to say the addition of fibres delayed crack propagation because fibres bridge the cracks at a micro-scale that controls the crack width and improves the stress distribution in the matrix at the time of loading, ensuring a better ductility that meets the need for improved tensile capacity due to the increased demands for strength and durability features due to the construction of large-scale infrastructure [15,16,17,18].

There have been numerous studies conducted on the use of coconut fibre in concrete and its effects on the mechanical properties of concrete. Some key findings from these studies include improved tensile strength, flexural strength, and enhanced impact resistance [19,20]. Majid Ali. et al. [21] investigated the influence of different fibre content and length on the damping ratio and fundamental frequency of simply supported beams when made with reinforced concrete with coconut fibres. They found that the damping of beams increased while their fundamental frequency decreased with structural damage. Beams with a higher fibre content experienced higher damping but a lower dynamic and static modulus of elasticity. 

Additionally, the use of coconut fibre has been shown to reduce the permeability of concrete, leading to improved durability and resistance to corrosion. Mahyuddin Ramli et al. [10] found that fibres helped prevent the development of cracks in concrete and could reduce the damage caused by aggressive environments. However, they indicate that the amount of coconut fibre used should be limited to 1.2% of the binder volume to avoid the negative effects of natural degradation.

Plastic shrinkage is a common phenomenon in concrete that occurs during the early stages of its curing process. This occurs when the surface of freshly poured concrete dries out faster than the interior due to the evaporation of water [22,23,24,25]. Consequently, the surface of the concrete contracts and shrinks, while the interior remains relatively unchanged. This differential shrinkage can result in cracking. This phenomenon not only compromises the structural integrity of the concrete but also affects its aesthetics. 

Plastic shrinkage is a significant issue in the construction industry and can be influenced by various factors such as temperature, humidity, wind velocity, and the mix design of the concrete [26]. Therefore, it is essential to understand the causes and effects of plastic shrinkage to ensure the durability of concrete structural elements, especially for surface elements such as slabs. Generally, these elements are prone to environmental attacks due to their extensive surface area, and as a result, shrinkage cracks commonly appear on their surfaces. 

In order to examine how shrinkage affects concrete slabs and to determine the impacts of incorporating fibres on cracking behaviour, it is important to recreate the environmental conditions in a laboratory with precise and controlled parameters. Generally, researchers used a wind tunnel for this simulation. Ghoddousi Parviz et al. [27] specified the preparation of 13 self-compacting concrete samples, which were made according to different mixing designs. These samples were placed inside a wind tunnel, which simulated warm and dry environmental conditions. A linear relation was observed between plastic shrinkage and the difference between the evaporated and evacuated water, which led to cracking. 

Sébastien Wolf et al. [28] used a wind tunnel to study the cracking behaviour of concrete slabs reinforced by steel fibres. They used two samples in each test set: a larger slab with dimensions of 160 × 60 × 8 cm^3^, which were anchored to prevent shrinking, and a smaller slab with dimensions of 30 × 30 × 8 cm^3^ used to measure water evaporation. Steel fibre reinforced concrete was poured into the slabs, and with a wind tunnel, the drying process was simulated. When the slab had been concreted, it was placed in an unfavourable wind tunnel for 7 h, following which the wind tunnel was removed to examine the cracking behaviour. By using their proposed testing approach, they were able to measure the width and length of cracks in the concrete slab elements, which allowed them to assess the impact of shrinkage on the slabs. 

This present work presents a new test methodology to study the cracking behaviour of concrete slabs using a hot air chamber with controlled environmental conditions. Cracks were created due to the concrete’s plastic shrinkage. In this way, it was possible to know whether there were improvements in concrete behaviour compared to conventional concrete with no fibres (coconut tow fibres added in their natural state). The crack length and crack depth were the parameters applied to evaluate the cracking behaviour of the concrete slabs reinforced by natural fibres. Finally, the proposed test methodology was suitable for the features studied to carry out a comprehensive experimental program.

## 2. Materials and Methods

Crack appearance is a problem that often happens upon concrete drying, especially in elements with large surfaces. Introducing natural fibres into concrete can help reduce crack propagation in such elements. In this study, coconut tow fibres were used in the concrete to evaluate their effect on cracking behaviour during the drying process at an early concrete age and under controlled environmental conditions. For that purpose, a hot air chamber (wind tunnel) was designed to perform an experimental test that could be applied to slab surfaces with the hot wind at a controlled speed and also a controlled temperature. With the proposed prototype, it was possible to obtain the evaporation rate of concrete’s water due to the hot air passed over the specimen’s surface, the cracking propagation process, crack length, and crack depth needed to evaluate the shrinkage behaviour of the concrete was reinforced by natural fibres.

### 2.1. Test Setup

#### Main Wind Tunnel Components

The wind tunnel design criteria were taken from the work of Wolf [28]. Those authors used a wind tunnel to study early-age shrinkage crack distribution in concrete plates reinforced with different steel fibre types. The cracking behaviour of concrete slabs with and without steel fibres was evaluated by this wind tunnel. 

A similar model for the proposed wind tunnel was modelled in SolidWorks to check the proposed model and observe the wind flow inside the tunnel, and check the accuracy of the dimensions. All the components were designed to provide the best efficiency and performance. The feasibility and functionality of the proposed model were evaluated by analysing the fluid flow inside the tunnel. After several modelling and related analyses, the final design shown in Figure 1 was achieved.

In order to induce environmental conditions in modelling, it was necessary to add a heater, a fan, and a sensor to measure both temperature and wind speed. These air chamber elements were placed by subdividing two blocks, as shown in Figure 2.

Figure 3 shows Block 1, where the spaces assigned to embed the fan and to place the heater for them to work together can be seen. It is worth mentioning that the fan had to lie behind the heater. This equipment location order was applied considering how the heat produced by the heater could harm the fan’s operation. To create a uniform wind flow, a set of square channels was placed in front of the wind outlet (see Figure 2a,b).

Figure 4 demonstrates Block 2, which is made up of two parts:A base that is fixed and firm, where the plate G mould, the anemometer, and the scale sit. It covers an area of 1600 mm× 700 mm.A mobile structure (lid) that facilitates placing the mixture inside moulds and avoids unnecessarily transferring plates to the base. This section is 400 mm high. It also allows a suitable distance between the camera and the plate to take images of the cracks that appear during tests.

### 2.2. Materials and Specimens

In order to evaluate the progressive fibre content reduction and its effect on slab cracking behaviour, and to compare it to concrete without natural fibres, G plates were made the concrete with 0% fibre, while other specimens were made by incorporating the three coconut doses (0.5%, 0.75 and 1%) into their mixture. A 2.5 cm slab thickness was chosen, and the selected diameter of the largest aggregate was 3/8″. It should be noted that the water-cement ratio (*w*/*c*) value was modified due to aggregate absorption, and also to previously immersed fibres in water for 24 h in the saturated surface dry (SSD) application state to avoid changes in the obtained *w*/*c* dose ratio.

For this research work, a group of specimens with fibre in its natural state was considered. To work with these fibres, the recommendation by Standard ACI 544.4R-18 [29] was applied, which states that fibres of vegetable origin should be cut to 4 cm.

Table 1 and Table 2 show the dose applied to the preliminary concrete design for the concrete without fibres (0%). Its average compression resistance came close to 200 kg/cm^2^. However, a decision was made to conduct this research with this dose because the average compression resistance in most small housing constructions in Ecuador was about 210 kg/cm^2^.

The cement used in this work was a product of the Holcim company (type GU, equivalent to cement type I) with a density of 2.90 kg/cm^3^. The properties for the characterization of the coarse and fine aggregates are provided in Table 2.

As shown in Table 3, for each concrete type (different fibre doses), four types of specimens were made: the main slab specimen, G plates (75 × 40 × 2.5 cm); a small slab specimen to monitor the water evaporation rate, P plates (30 × 30 × 2.5 cm); concrete cylinder specimens (10 × 20 cm); and prismatic beam specimens (10 × 10 × 40 cm) to obtain flexural strength. Figure 5 shows the employed specimens corresponding to each type.

Table 4 presents the designed mixture of the investigated fibre-reinforced concrete at 0.5%, 0.75%, and 1% fibre doses. For measuring the workability of freshly mixed concrete, a slump test was performed on each batch of freshly mixed concrete to assess its workability. Concrete batches containing 0%, 0.50%, 0.75%, and 1% of coconut fibres were tested, and the average slump values obtained were 5.1 cm, 2.8 cm, 1.2 cm, and 0.0 cm, respectively. 

The results indicate that as the proportion of coconut fibres in the concrete increased, the workability decreased. The batches containing 0.75% and 1% coconut fibres had significantly lower slump values which indicated poor workability and difficulty in handling and placing the concrete. One possible solution for addressing this issue was to utilize a superplasticizer in the concrete, which could enhance its workability. In a study by Syed et al. [30], different levels of coconut fibres (0%, 0.6%, and 1.2%) were incorporated into the concrete, and a superplasticizer was employed to increase the slump value. The researchers used varying amounts of the superplasticizer (75, 195, and 370 mL) for the aforementioned concrete mixtures, respectively. This approach was found to effectively improve workability, especially for concrete with a high fibre content, and the slump values for concrete with 0%, 0.6%, and 1.2% coconut fibres were measured at 15, 20, and 50 mm, respectively.

### 2.3. Instruments

Anemometer

The anemometer (Figure 6) is a sensor that allows the temperature and wind speed produced inside the hot air chamber to be measured and recorded. When a fan is placed inside the wind tunnel, it is possible to adjust and change the wind speed to simulate different climate conditions. Therefore, it was necessary to include a sensor to monitor wind speed during the test process. 

This equipment consists of a propeller that measures airflow, a handle with a temperature sensor, and a screen that collects information from the sensor. The wind speed range that this instrument can measure is 0.91–44.99 m/s, and temperatures could range from 0 °C to 59.99 °C. 

2.Heater

In order to produce hot air, it was necessary to purchase a strong industrial air heater with power levels of 3000/4000/5000 watts, which heated the wind created by the fan. This equipment (Figure 7) has adjustable louvres to better distribute the heat wind flow, an adjustable thermostat, and a system to prevent overheating the equipment.

3.Fan

For the airflow to reach the plates’ surfaces, it was necessary to use an electric fan (Figure 8) to push hot air, pass it through the grid (transforming the airflow into laminar air), and move it to the study area where the plates were. For this purpose, a 3-speed fan was used.

4.Digital Balance

In order to determine moisture reduction on the concrete surface of the P plates, a digital weighing balance with a maximum capacity of 40 kg was used (Figure 9). With this weighing balance, any variation in concrete mass during the testing process was recorded (6 h). These data were recorded every 15 min. It is worth mentioning that to measure these data, a wooden mould was used to cast the P plates, which were placed on the balance. 

5.Camera

In order to monitor the cracking process on the concrete plate surfaces, it was necessary to use a high-resolution camera to record crack formation every 15 min during the 6 testing hours. Each picture records the cracks on the G plate surfaces during the testing time and can be used to determine the crack length according to the details in Section 2.3.2. 

6.Ultrasound equipment.

This device (Figure 10) helps to know the crack depth. This test was performed 24 h after manufacturing the G plates. This device generated longitudinal waves that went from an emitter to a receiver (transducers). These waves are not audible because their frequency is below 20 Hz (infrasonic waves), and they travel through concrete. This means that the test is non-destructive and can be applied at any time, provided that concrete is in a hardened state. This process is shown in further detail in Section 2.3.3.

#### 2.3.1. Crack Pattern Monitoring

In order to record crack formation on the surface of concrete, pictures were taken during the 6 h testing period with plates inside the hot air chamber, which were subjected to controlled environmental conditions. After 18 h, they were left outside the test chamber.

All the pictures were taken in the same position and at the same height to ideally visualise each G plate section.

#### 2.3.2. Crack Length Measurement

In order to determine the crack length in the G plates, it was necessary to distinguish the appearance of cracks in two groups. The cracks that formed during the 6 h testing were painted blue, while those that occurred outside the chamber, namely at 18 h of testing, were painted red (see Figure 11).

In order to simplify the measurement process, a picture was taken of the entire plate. It was scaled in AutoCAD and delimited by areas, as shown in Figure 11. Polylines were drawn to obtain the total crack length sum and, thus, to record the data for their respective analysis.

#### 2.3.3. Crack Depth Measurement

The crack depth measurement procedure was conducted by extending the distance between the ultrasound transmitting and receiving sensors on the crack, with which the position was determined from the equipment’s lamp and buzzer (see Figure 12).

If it was not possible to place sensors at the same distance from a crack, the depth of the crack could be calculated using Equation (1), with the parameters a and b indicated in Figure 13.
(1)$$y=\sqrt{a*b}$$

It is worth mentioning that the ultrasound equipment has a limitation in crack depth measurement terms because it can only measure depths that exceed or equal to 1.50 cm.

### 2.4. Procedure

Based on the mixtures designed for concrete with and without fibres, concrete was prepared, and the G and P plates were cast in moulds that were located inside the wind tunnel. To avoid retaining air bubbles, the concrete was compacted with a smooth rod. Finally, the plates’ surface roughness was removed to well visualise cracks on the surface and also to obtain optimal photographic records for the analysis. The concrete mixture was mixed in a mixing machine, and fibres were added to the mixed concrete when it had been well mixed.

The wind was induced by using the fan. Level 2 was determined as suitable for maintaining wind speed within the 1.20~1.34 m/s range. To produce hot wind waves, a heater thermostat was placed at the highest level, and a 5000-watt power switch was used. With these specifications and those of the fan, a maximum temperature of 60 °C was achieved. With the help of the anemometer, these data were recorded during the 6 h testing period over 15 min periods.

After a period of 6 h, the hot wind stopped. The samples in the wind tunnel were not removed, but cracks were marked with blue ink and exposed to laboratory environmental conditions. The following day, after the specimens had been left outside the chamber for 18 h, new cracks appeared and were labelled with red ink. Plate G, the main sample, was then taken out, and an ultrasound examination was performed.

During the test performance, the mass of the P plate was recorded every 15 min during the 6 h testing period. Then, the plate was removed from the chamber. The next day, its weight was recorded again.

## 3. Test Results and Discussion

### 3.1. Crack Lenght

It is known that cracks are a problem in concrete elements because they affect both their aesthetics and their durability if cracking is excessive. For this reason, plates (G) with three fibre addition percentages 0.5%, 0.75%, and 1% were analysed to determine the total crack length that resulted from accelerated drying and also the crack depth. Table 5 shows the crack length values obtained from plate G after the 6 h testing period inside the hot air chamber and after 18 h of being left outside it.

As Table 5 indicates, the total crack length notably decreased due to the incorporation of fibre compared to conventional concrete. The crack length decreased in relation to the increasing percentage of fibre. The greatest reduction occurred from 1% coconut fibre to 95% in relation to the plate with no fibre addition. For the 0.5% and 0.75 percentages, crack reductions of 54.43% and 72.40% were also significant. 

By incorporating coconut fibres into the concrete matrix, it was evident that the total number of appeared cracks on the surface of the slab decreased significantly from 252.06 cm to 80.42 cm when compared to concrete without fibres and concrete with 0.75% fibre content. This decrease in the number of visible cracks was crucial for enhancing the durability of the slab elements and providing better protection against environmental factors. Based on the results obtained, it can be concluded that the use of coconut fibres had a significantly positive effect on the cracking behaviour of surface elements, resulting in fewer and shorter appearances in the cracks, which, in turn, improved both the durability and aesthetics of the concrete element.

According to these results, it can be concluded that the best option for good crack length reduction purposes is the 1% coconut fibre addition for the coarse aggregate mass because this fibre acts as a coarse aggregate replacement. However, it should be mentioned that the slump would significantly reduce, and its workability would diminish.

### 3.2. Crack Depth

It is important to evaluate crack depth because deep cracking can cause the entry of aggressive agents such as chlorides, carbonates, or sulphates, which can cause steel reinforcement corrosion and concrete carbonation. All this can reduce the reinforcement elements’ durability, especially for slab members. Accordingly, to evaluate both crack depth and the fibre volume content influence on this phenomenon, two concrete types were studied, namely a plate with 0% fibre and another with 1% fibre, to compare their depth cracking behaviour in relation to coconut fibre addition. The monitored cracks were numbered from 1 to 10 (F1 to F10), as indicated in Table 6, and depth was measured by ultrasound equipment.

As observed from the data in Table 6, the maximum crack depth for the specimen with 0% fibres was 2.5 cm. This depth size was the same as the plate thickness, which meant that the cracks were able to cross the entire plate depth. For the specimens with 1% fibre, the maximum depth was 1.5 cm. It can be concluded that the addition of fibres (1% in this case) offered significant improvements in reducing crack depth. It should also be noted that the specimen with 1% fibres was that for which most of the appeared cracks were less than 1.50 cm (minimum depth measured by ultrasound equipment). This indicates that fibres were able to withstand the stresses caused by concrete shrinkage and, as a result, their crack number and crack depth decreased.

### 3.3. Moisture Loss

After the drying hours (due to the wind passing over the sample) had passed, the moisture loss in the P plates decreased. This caused concrete hardening and a visible surface drying state. Mass reduction (G plate mass) due to moisture loss was approximately 0.01 kg in the first hour until the temperature stabilised. After this period, mass reduction decreased from 0.02 to 0.025 kg (measured every 15 min.). Figure 14 shows the moisture loss rate during the test time for the P plates as percentages.

The specimens’ moisture loss values were cumulatively presented, i.e., after 6 h of testing at 60 °C and at a wind speed of 1.34 m/s, where the specimens’ maximum moisture percentage went from 125% to 165%. The highest moisture loss rate was for the specimen with 0.5% fibres.

The external moisture loss of concrete refers to the loss of water from the surface of the concrete due to environmental factors, such as heat, wind, and low humidity. As concrete cures, it goes through a process called “surface evaporation,” where the water at the surface of the concrete evaporates into the air. If the rate of surface evaporation is too high, it can lead to a variety of issues such as cracking, crazing, and weakened surface strength [31]. Ziari H. et al. [26] studied the effects of temperature, relative humidity, and wind speed on plastic shrinkage cracking on concrete pavement. They used practical characteristics, such as temporal and geometric characteristics, to evaluate the behaviour of the cracking in different environmental conditions. The results showed that each environmental factor had a significant effect on the behaviour of cracking, with relative humidity having the most significant effect. Additionally, relative humidity had the most influence on both the temporal and geometric characteristics, and the effects of temperature and wind speed were close to each other.

### 3.4. Mechanical Properties

Table 7 shows the compressive strengths that were obtained from the tests performed on the cylinder specimens and the flexural strength of the beams subjected to bending after the 28-day curing time. In general, research indicates that concrete composite beams that incorporate natural fibres exhibit superior flexural strength and deflection resistance when compared to conventional concrete beams [32,33].

The effectiveness of coconut fibre in improving the compressive strength of concrete depends on several factors, such as the type and amount of coconut fibre used, the mix design of the concrete, the curing conditions, and the testing method used to measure the compressive strength.

The obtained compressive strength results demonstrate that increased fibre content favoured this mechanical property; an 11.06% increase was evidenced by adding 0.5% fibre and one of 2.39% with 0.75% compared to the concrete with no fibre (0%). However, with 1% fibre, compressive strength was lowered by 8.93%. This could be because adding too much coconut fibre would not allow good materials’ homogenisation, and a mixture with low resistance would form.

In general, the addition of a small amount of coconut fibre has been found to increase the compressive strength of concrete. However, adding too much coconut fibre can result in a decrease in compressive strength due to the formation of voids and other defects in the concrete.

Abdullah, A. et al. [34] investigated the effect of coconut fibre content on mechanical physical and mechanical properties of concrete. Coconut fibre was added as a reinforcement and replaced the composition of sand. Composites were developed based on 3 wt.%, 6 wt.%, 9 wt.%, 12 wt.%, and 15 wt.% of coconut fibre by mixing and curing process. The results showed that the compressive strength increased with the increasing amount of coconut fibre. The composite cement reinforced with 9 wt.% of coconut fibre gave the highest compressive strength value of 43.84 MPa after 28 days of curing, while the lowest value was 26.04 MPa for the composite reinforced with 15 wt.% of coconut fibre.

Moreover, the quality of coconut fibre used also plays a significant role in determining its effect on compressive strength. Coconut fibres that are properly processed, cleaned, and dried before adding to the concrete mix can yield better results.

Flexural tensile strength is the measure of a material’s ability to withstand bending forces. The flexural tensile strength of a material is determined by conducting a flexural test. The test involves applying a load at the centre of a sample supported at two points. The load creates a bending moment that causes the sample to bend, and the maximum stress at the bottom of the sample is measured. The flexural tensile strength is the maximum stress the sample can withstand before breaking and can be obtained with the following equation:(2)$$R=\frac{\mathrm{PL}}{{\mathrm{bd}}^{2}}$$

The Ecuadorian standard, NTE INEN 2554, 2011 [35], was used to conduct the flexure test, and the obtained results are provided in Table 8.

As can be observed, as the percentage of coconut fibre composition increased from 0% to 0.75%, there was an increase in the maximum applied load and the average flexural tensile strength. However, for the composition of 1.00%, there was a decrease in both the maximum applied load and the average flexural tensile strength. Therefore, the optimal composition for maximum flexural tensile strength appeared to be 0.75% coconut fibre. However, it is important to note that these results may be specific to the testing conditions and should be validated by further testing before making any practical applications.

Fibre length and orientation are important factors that affect flexural strength. Hwang et al. [36] conducted a study on the impact of adding short coconut fibres (length of 17mm) to cement composites with varying fibre volumes and water-to-binder ratios. The composites were then tested for mechanical properties, plastic cracking, and flexural strength. The study found that increasing the percentage of short coconut fibres (at 0%, 1%, 2.5%, and 4%) led to a significant increase in flexural strength, with a maximum increase of 44% seen in the samples with 4% coconut fibre. However, when using longer coconut fibres (length of 40 mm), our experimental results showed a decrease in flexural strength when the fibre content exceeded 0.75%. To gain a better understanding of this effect, further studies should be conducted with a wider range of fibre volume content.

## 4. Summary and Conclusions

This research proposes a new test methodology to evaluate how coconut fibres affected crack propagation rates resulting from plastic shrinkage on concrete slab surfaces. Tests were conducted using three different doses of natural coconut fibre (0.5%, 0.75%, 1%) to verify the validity of the equipment and methodology for evaluating slab member cracking behaviour. With the proposed wind tunnel design, climate conditions such as wind speed and air temperature were simulated in the laboratory. The study shows that using 0.75% coconut fibre in concrete slabs reduces plastic shrinkage cracks and improves mechanical properties.

Using 0.75% coconut fibre in concrete results in a 72.40% reduction of the crack length, also increased compressive strength by 2.39% and flexural tensile strength by 6.53%, compared to conventional concrete without fibre. 

The results obtained from the proposed test method demonstrate the suitability of this test method to study the cracking behaviour and plastic shrinkage effects of slab elements under controlled climate conditions. Consequently, the proposed test methodology is appropriate for future studies.

## Figures and Tables

**Figure 1 materials-16-03247-f001:**
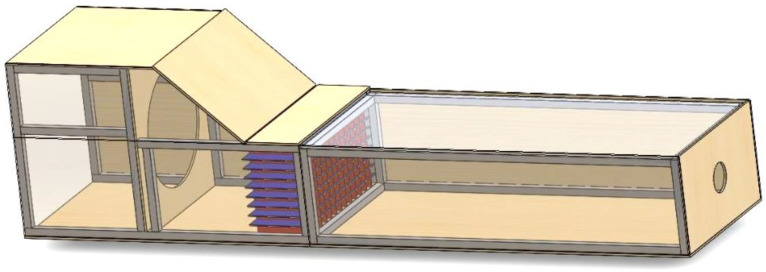
A 3D view of the proposed wind tunnel.

**Figure 2 materials-16-03247-f002:**
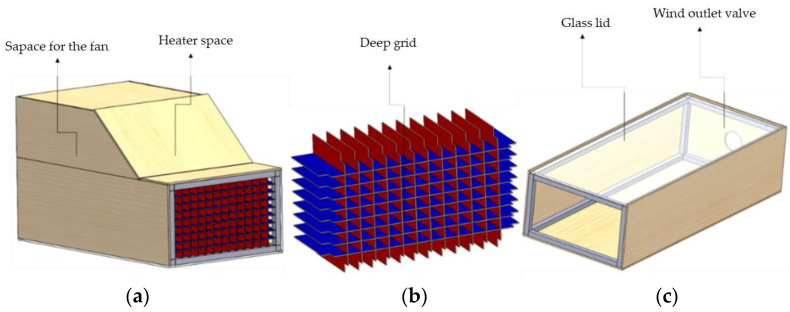
Hot air chamber components: (**a**) Block 1; (**b**) Deep grid; (**c**) Block 2.

**Figure 3 materials-16-03247-f003:**
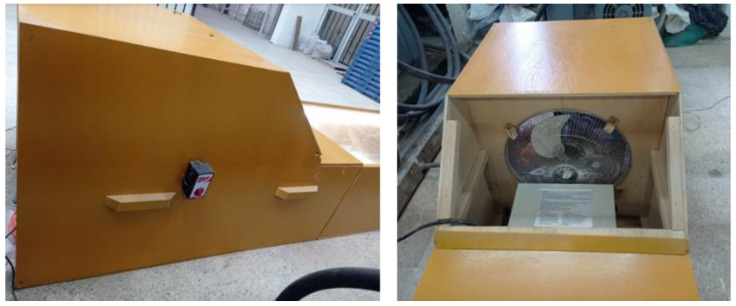
Block 1.

**Figure 4 materials-16-03247-f004:**
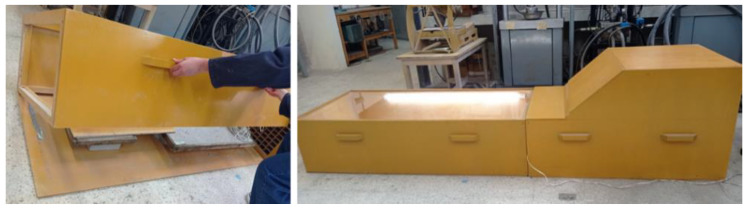
Block 2.

**Figure 5 materials-16-03247-f005:**
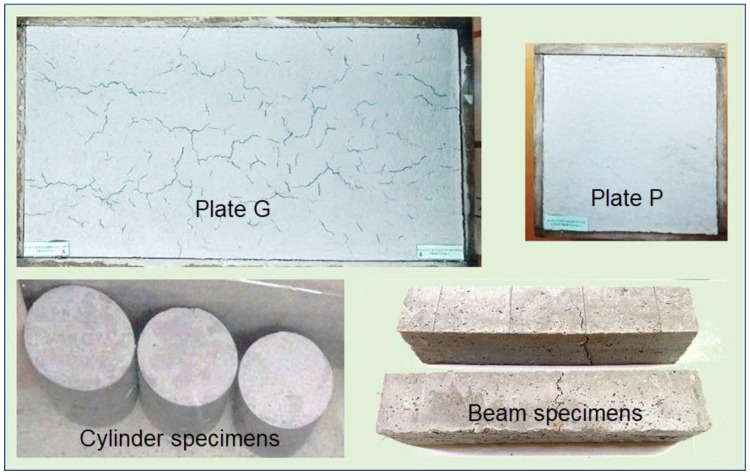
Type of test specimens used in the project.

**Figure 6 materials-16-03247-f006:**
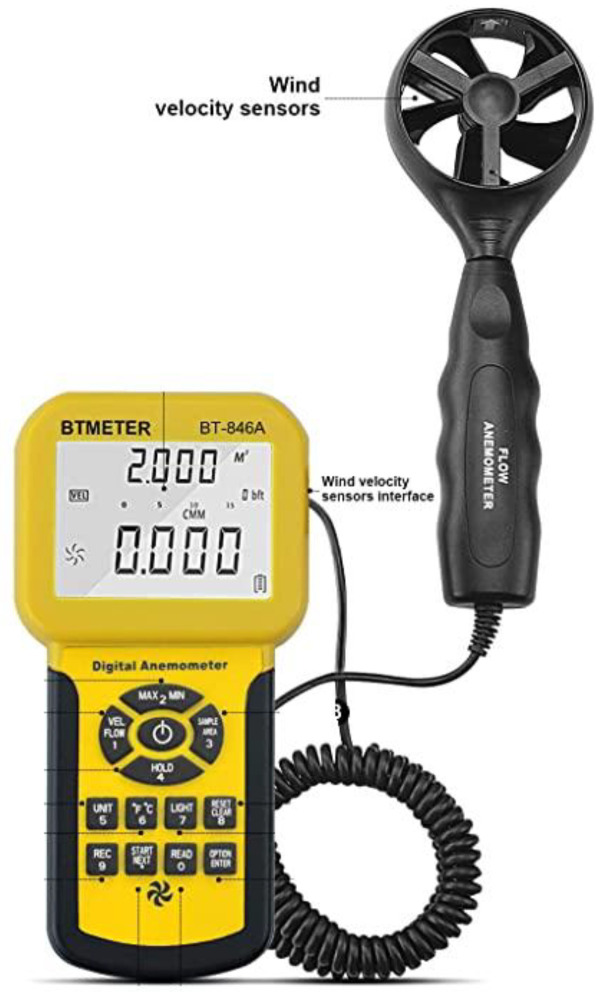
Anemometer: temperature and wind speed sensor.

**Figure 7 materials-16-03247-f007:**
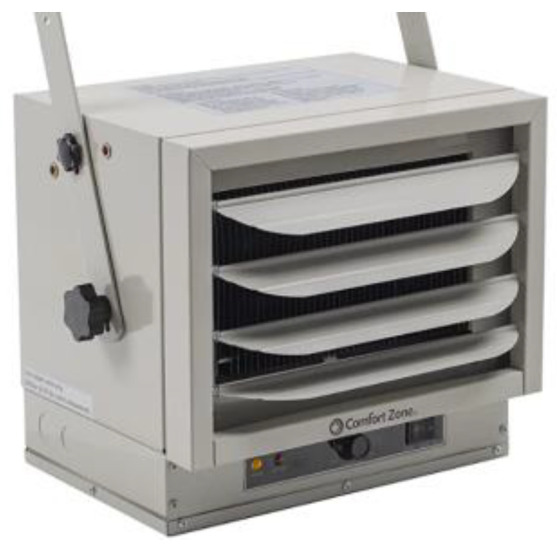
Industrial air heater.

**Figure 8 materials-16-03247-f008:**
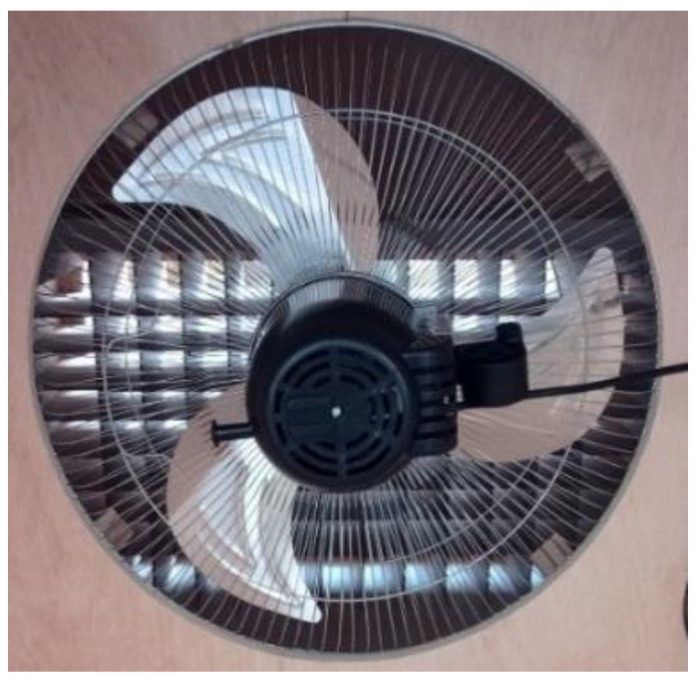
Oscillating 3-speed fan.

**Figure 9 materials-16-03247-f009:**
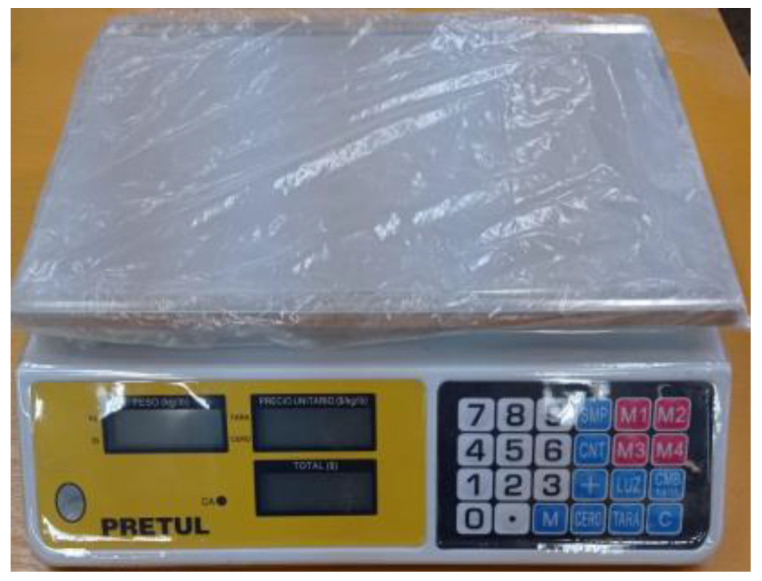
Digital scale with a 40 kg weighing capacity.

**Figure 10 materials-16-03247-f010:**
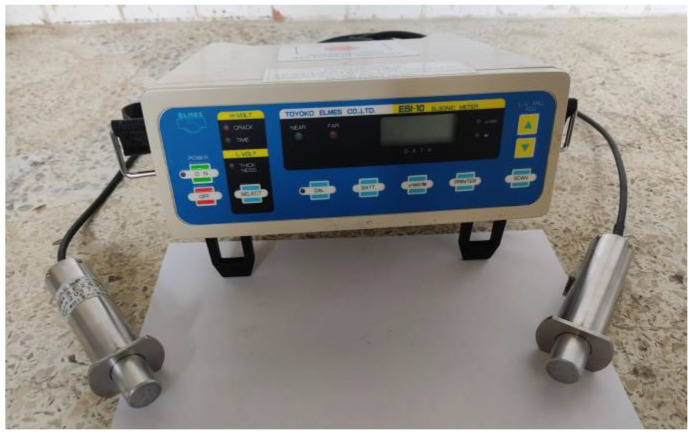
Ultrasound equipment.

**Figure 11 materials-16-03247-f011:**
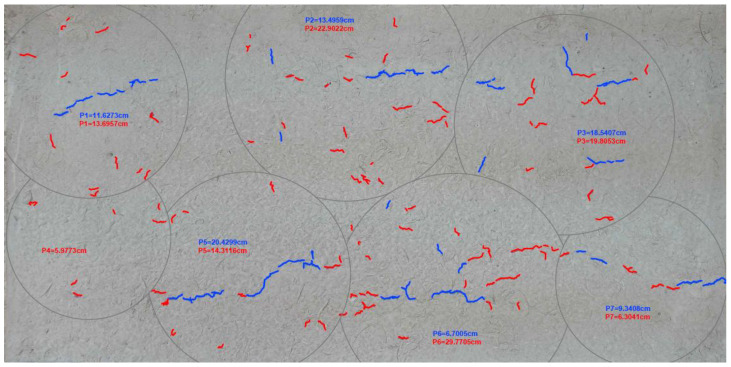
Distinguishing cracks on plates.

**Figure 12 materials-16-03247-f012:**
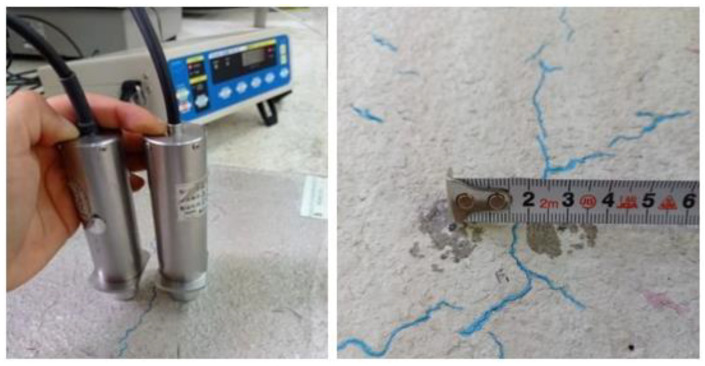
The crack depth measurement process using an ultrasound device.

**Figure 13 materials-16-03247-f013:**
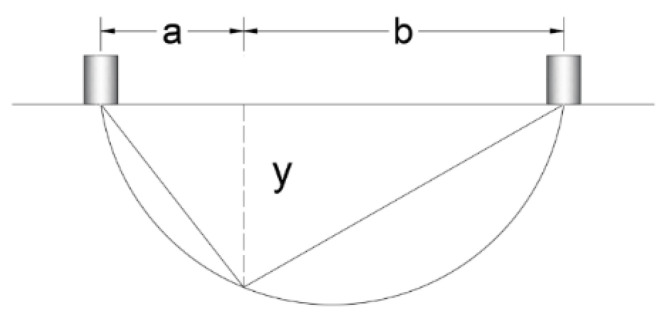
Graphical representation of Equation (1).

**Figure 14 materials-16-03247-f014:**
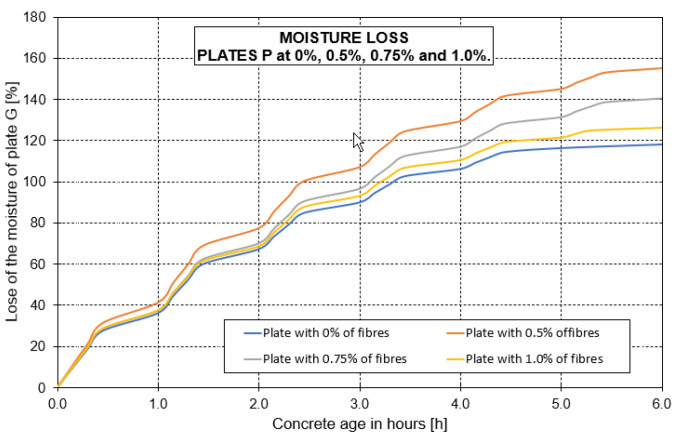
Moisture loss rate.

**Table 1 materials-16-03247-t001:** This dose with 0% stock for f′c = 200 kgf/cm^2^, coarse aggregate 3/8″.

*w*/*c*	*c*/*c*	Fa/c	Ca/c
0.52	1.00	2.09	1.46

c = cement; w = water; Fa = fine aggregate; Ca = coarse aggregate.

**Table 2 materials-16-03247-t002:** Coarse and fine aggregate characterization summary.

Properties	Coarse	Fine	Units
Fineness Modulus [FM]	4.05	2.96	-
Specific Density [SD]	2.60	2.61	gr/cm^3^
Apparent Specific Density [ASD]	2.77	3.02	gr/cm^3^
Water Absorption	2.43	1.53	%
Bulk Density	1.27	1.76	gr/cm^3^
Uniformity Coefficient	0.25	2.53	-

**Table 3 materials-16-03247-t003:** Number of specimens contemplated in the project.

Coconut Fibre Percentage	Daily Assay		Assay at 28 Days	
	G Plates	P Plates	Beams	Cylinders
0.00%	1	1	3	3
0.50%	1	1	3	3
0.75%	1	1	3	3
1.00%	1	1	3	3

**Table 4 materials-16-03247-t004:** Mix proportions of concrete.

Water [kg/m^3^]	271.21
Cement [kg/m^3^]	438.46
Sand [kg/m^3^]	917.35
Gravel [kg/m^3^]	638.00
Coconut fibre [%]	0.50, 0.75, 1.00

**Table 5 materials-16-03247-t005:** The 6 h and 18 h crack lengths.

Coconut Fibre Percentage Composition (Vf)	6 h Inside the Chamber [cm]	18 h Outside the Chamber [cm]	Total Crack Length [cm]	Percentage Based on Vf = 0%
0.00%	370.08	252.06	622.14	0.00%
0.50%	217.43	66.06	283.49	54.43%
0.75%	91.28	80.42	171.70	72.40%
1.00%	8.02	17.26	25.29	95.93%

**Table 6 materials-16-03247-t006:** Crack depth measured by ultrasound equipment (units: cm).

Cracks	1%	0%
F1	1.50	1.90
F2	X ^1^	1.73
F3	X	1.83
F4	X	1.67
F5	X	1.81
F6	X	1.97
F7	X	2.50
F8	X	1.70
F9	X	2.50
F10	X	1.50
Max. depth	1.50	2.50
Min. depth	1.50	1.50

^1^ “X” represents measurements below 1.5 and, thus, cannot be considered “1.5”.

**Table 7 materials-16-03247-t007:** Compressive strength of cylinders and flexural tensile strength of beams.

Coconut Fibre Percentage Composition	Compressive Strength [kgf/cm^2^]
0%	201.87
0.50%	226.97
0.75%	206.82
1.00%	183.85

**Table 8 materials-16-03247-t008:** Flexural tensile strength of beam test results.

Coconut Fibre Percentage Composition	Maximum Applied Load [KN]	Average Flexural Tensile Strength [MPa]
0%	11.95	3.35
0.50%	13.92	3.90
0.75%	15.93	4.30
1.00%	12.72	3.53

## Data Availability

https://bibdigital.epn.edu.ec/handle/15000/23329.

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
