# Peer review of "Test Method for Studying the Shrinkage Effect under Controlled Environmental Conditions for Concrete Reinforced with Coconut Fibres"

_materials, 2023, doi:10.3390/ma16083247_

Round 1

Reviewer 1 Report

This paper presents an experimental study to investigate the shrinkage effect on concrete slabs reinforced with coconut fibres under controlled environmental conditions. The crack length was determined to evaluate the fibre content effect on the crack propagation on slab surfaces. The results show that the concrete reinforced with 0.75% fibre content exhibits a reduction of crack propagation on slab surfaces and an enhancement of flexural and compressive strengths.

This research work is interesting; however, it should be enhanced taking into account of the following comments.

1- In Practical (in field), where these coconut fibres reinforced slabs could be used?

2- Page 3, line 115: Cite the name of authors in the following sentence “Teixeira and Andrade investigated….”.

3- The literature review should be improved adding some results published in previous studies [23-28].

4- Page 5, line 200: The symbol a/c should be defined for the first in the text.

5- How the mix proportions of concrete were obtained?

6- The test procedure, in the subsection 2.4, is not clear and should be more reformulated (line 322 to 326). G plate specimen was placed inside the chamber for 6 hours and not after 6h!!

7- Page 9, line 301: correct the following sentence “If it is impossible to spread sensors…”

8- Page 10, line353: correct the following sentence “two concrete types were studied…”.

9- What about the durability of the coconut fibres reinforced concrete?

10- Complete the authors names in the reference list such as in reference 1, 2, 3, 4, …etc.

Reviewer 2 Report

1. In the abstract, please check the second sentence and rewrite it. “A wind tunnel was designed to simulate climate……..” Why this statement came suddenly?

2. Abstract has to be well-written to include the problem statement, the bigger picture of the work done, and the important result and findings of the paper, kindly rewrite it.

3. What is the significance of wind tunnel test on the slab?

4. Introduction part doesn’t have a sequential order and flow. Kindly rewrite

5. Authors may consider citing the following articles in the introduction part:

a) https://doi.org/10.14419/ijet.v7i3.12.15906

b) https://dx.doi.org/10.3923/ajaps.2014.232.239

6. Novelty of the work has to be justified properly, since plenty of works were done with concrete slabs reinforced with coconut fibre from 1980 onwards.

7. Kindly include the fresh properties in the work.

8. How do you justify the applicability of structural slab members with 2.5 cm thickness?

9. In mix proportion, the coarse aggregate content was nearly half of fine aggregate, how do you justify such proportion for concrete slab members? Any standards available?

10. In the table of crack depth, it was found that concrete slab with 0% fibre has a crack depth of 2.5 cm does it mean slab has a crack on entire depth?

11. Author may include some test results of concrete slab without being exposed to wind tunnel and may include a comparative study

12. For conclusion, avoid long sentences and put it in point wise.

Reviewer 3 Report

Regarding this manuscript I think that is potentially acceptable for publication, but I suggest a major revision with the following clarifications and modifications:

- You must include a new table with the detailed concrete dosage in kg/m3 (weight of the components, no proportions or percentages).

- You should include the more relevant physical and mechanical properties of the cement, gravel, sand, etc. Comments about the additive used

- The experimental research is very small. Perhaps, you can add more information...

- A relevant property of the fibres concretes is the flexural strength. I think that you must include results about the evolution of this parameter regarding the percentage of substitution.

 - In the discussion of the results, you should compare them, with those obtained in other studies in all the properties analysed. You only make it in some properties

Round 2

Reviewer 1 Report

All comments are carried out and the paper is enhanced and can be published.

Author Response

Great news! We appreciate the time you took to review our article, and all of your comments were helpful in improving our work. Thank you.

Reviewer 2 Report

Comments were carefully taken care.

Author Response

(The authors gave the same response as above.)

Reviewer 3 Report

The authors have not ansewred to the following questions:

A relevant property of the fibres concretes is the flexural strength. I think that you must include results about the evolution of this parameter regarding the percentage of substitution.

In the discussion of the results, you should compare them, with those obtained in other studies in all the properties analysed. You only make it in some properties

Author Response

We greatly appreciate all your valuable comments. We have revised our manuscript, according to the reviewers’ comments, questions and suggestions. We believe that the manuscript has been further improved. You will find the answers to your questions in the attached document.
